# Design of Concurrent Tri-Band High-Efficiency Power Amplifier Based on Wireless Applications

**Mingming Gao, Gaoyang Xu *** and **Jingchang Nan**

School of Electronic and Information Engineering, Liaoning Technical University, Huludao 125100, China
* Correspondence: xugaoyang2020@163.com; Tel.: +86-18160748423

**Abstract:** To meet the existing requirements of multiband communication and improve the efficiency and performance of communication RF modules, a concurrent tri-band high-efficiency power amplifier operating in three frequency bands is proposed. The input and output impedance values of concurrent power amplifier is analyzed, and the input and output-matching circuit and bias circuit are designed. Through the impedance compensation principle, the impedance matching of three frequency bands is realized, and the amplifier can maintain high power and high efficiency at three arbitrary wide interval frequencies. To this end, a simultaneous tri-band power amplifier is designed and tested by using transistor CGH40010F. The experimental results show that the peak power of the designed simultaneous tri-band high-efficiency power amplifier is more than 10 W, the power-added efficiency reaches 55~69%, and the amplification gain is about 10 dB at three frequency bands of 2.2, 2.6, and 3.5 GHz. The design of concurrent tri-band high-efficiency power amplifier is flexible, the calculation of microstrip line parameters is simple, and it can work in three frequency bands simultaneously. It provides an effective structure scheme for designing concurrent power amplifiers in transmitting systems.

**Keywords:** concurrent tri-band; impedance compensation; impedance matching; high efficiency; power amplifier





## 1. Introduction

With the rapid development of modern wireless communication technology, the number of base stations demand surges. The power amplifier is one of the most important devices in the receiving and transmitting module of the base station, and it is also a hot topic in communication research [1]. The power amplifier is a large power consumption device. It is a difficult problem to optimize the performance of the power amplifier. The design of a multiband concurrent power amplifier can solve the problem of spectrum resource shortage and matching failure in multi-frequency signal modulation. Therefore, concurrent multiband power amplifiers (PA) are also considered one of the possible solutions for modern and future communication systems [2]. In recent years, dual-band concurrent power amplifiers have been reported, such as tri-band or quad-band concurrent power amplifiers, have been proposed. Andres et al. [3] proposed a D/S class power amplifier with concurrent tri-band based on small base stations, and Li et al. [4] designed a Doherty power amplifier with concurrent quad-band using *T*-type coupling lines.

In the early research of concurrent multiband power amplifiers, Rawat et al. [5]. in India proposed a $\pi$ and *T*-type concurrent dual-band matching circuit structure in 2011, which can match any arbitrary reflection coefficients seen at two unrelated frequencies with standard 50 Ω loads. The $\pi$ and *T* structures are equivalent to a quarter wavelength microstrip line, and the theoretical derivation is carried out in detail. However, there is a problem with the design. For impedance matching, sometimes only a *T* or $\pi$ structure is used, and the solution of the equation does not produce a valid solution. At this time, another $\pi$ or *T* structure is added to the literature, and the output impedance value is

first transited to an intermediate impedance value, and then a $\pi$ or $T$ structure is used to match the impedance to 50 $\Omega$ to achieve the impedance matching of the dual-band. In 2015, Wu et al. [6] proposed and discussed a novel dual-band matching network for parallel dual-band power amplifiers. The first stage matching network uses transmission lines and short circuit and open-circuit transversals to effectively convert from real impedance to complex impedance. However, the electrical parameter values are obtained by adjusting and observing the Smith diagram, and no analytical design method is given. In the same year, Fu et al. [7] introduced the design of a new flexible two-frequency band complex load impedance transformer. The proposed dual-band impedance converter can match any complex impedance of the active device at two arbitrary frequency bands to a load impedance of 50 $\Omega$. At the same time, the closed-form Equation is analyzed and derived to obtain the realizable solution for any frequency band ratio. During this period, many concurrent multi-frequency band power amplifiers [8–10] have been studied successively. In 2021, Zhang et al. [11] proposed a concurrent dual-band power amplifier using a coupler and used the dual-band coupler to design the output circuit of the power amplifier. Moreover, the harmonics are controlled, so the test results show that the power and efficiency are relatively suitable. The harmonic tuning technique is an effective method to achieve high-efficiency power amplifiers. It is also used in the design of the dual-bandpower amplifier. In the same year, Pang et al. [12] designed a dual-band high-efficiency power amplifier based on harmonic tuning. A new system synthesis method for dual-band matching networks is presented. In the first step, the target complex impedances at two operating frequency bands are converted into real impedances. The second step is a dual-band filter to ensure that the real impedance of the previous stage matches the terminal impedance. A transmission line is added between the two stages to adjust the impedance of the second and third harmonics without changing the impedance at the fundamental frequency. Harmonic control is very effective for improving the efficiency of RF transistors, especially when using the Class-F [13–17] work model.

To realize the concurrent operation of power amplifiers in more frequency bands, this paper proposes a concurrent tri-band high-efficiency power amplifier. The impedance compensation principle is adopted. The input, output, and bias circuits of the concurrent tri-band high-efficiency power amplifier are designed, and the input and output impedance matching of three frequency bands and the high isolation between each band are realized. The formula is deduced in detail. The design of the power amplifier is simple, and the circuit parameters are solved conveniently. To verify the feasibility of the scheme, a concurrent tri-band high-efficiency power amplifier operating at 2.2/2.6/3.5 GHz is designed. The test results show that the power amplifier has a peak output power of more than 10 W, a power-added efficiency of 55~69%, and an amplification gain of about 11 dB in all three frequency bands.

## 2. Design Method of Matching Circuit

### 2.1. The First Band Matching Circuit Design

Firstly, three frequency bands are measured by load pull $f_1, f_2, f_3(f_1 < f_2 < f_3)$ The complex impedances of are, respectively, $Z_{L1} = R_1 + jX_1$, $Z_{L2} = R_2 + jX_2$, $Z_{L3} = R_3 + jX_3$ The overall output-matching circuit is shown in Figure 1. According to Figure 1, the characteristic impedance of a series microstrip line in a matching circuit is fixed as $Z_0 = 50\ \Omega$, And the real part $R_1$ of the admittance of the corresponding frequency band is transformed to $G_0 = 0.02$, The first microstrip parallel branch TL1 can transform the imaginary part $X_1$ of the impedance of the first frequency band $f_1$ to zero. At point A, the impedance is realized in the first band $Z_{L1} = R_1 + jX_1$. Match to load $Z_{load} = 50\ \Omega$. The characteristic impedance of the series microstrip line TL3 in the second section is $Z_0 = 50\ \Omega$. Transform the real part of the admittance of the second frequency band $f_2$ to $G_0 = 0.02$. At point H, the second $T$-type microstrip line structure realizes the imaginary part $X_4$ of the second frequency band $f_2$ transformation to 0. The impedance $Z_{L2} = R_4 + jX_4$. Transform to $Z_{L2} = 50\ \Omega$. At the same time, it is equivalent to an open circuit at the first

frequency band $f_1$. The characteristic impedance of the third series microstrip line TL7 is $Z_0 = 50\ \Omega$. The real part $R_6$ of the admittance of the third frequency band is transformed to $G_0 = 0.02$. At point N, the third $\pi$-type microstrip line TL8, TL9, TL10, TL11, and TL12 structure realizes the imaginary part $X_6$ of the third frequency band transformation to 0, The impedance $Z_{L3} = R_5 + jX_5$. Transform to $Z_{L3} = 50\ \Omega$. At the same time, the third $\pi$-type microstrip line for the first and second frequency bands $f_1$, $f_2$ are equivalent to open circuits. Finally, the impedance matching of three frequency bands is realized.

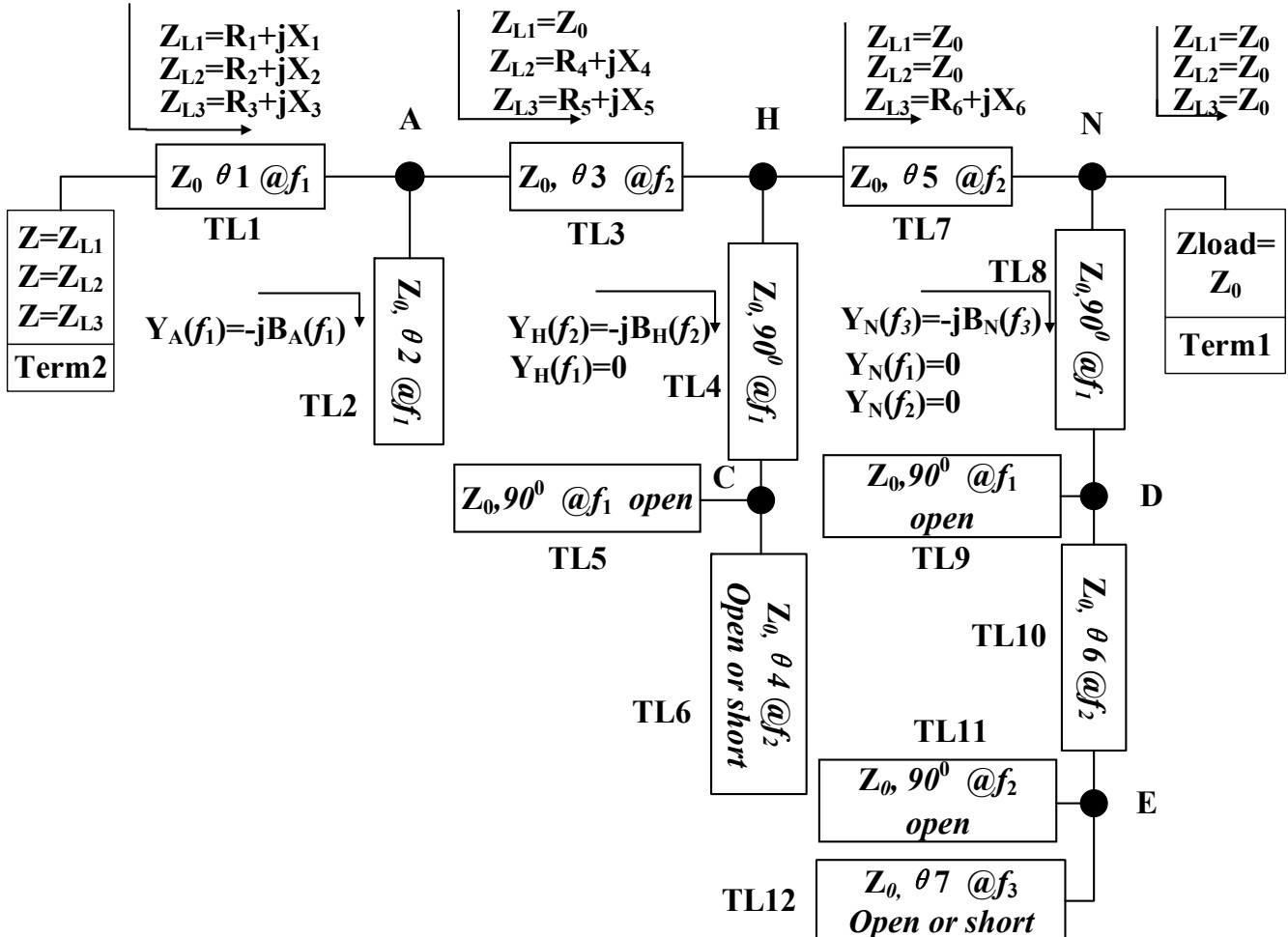

**Figure 1.** Match circuit structure diagram. The default is $Z_0 = 50\ \Omega$.

The input impedance at the first frequency band is:

$$Z_{L1} = R_1 + jX_1 \tag{1}$$

The input impedance equation of the series microstrip line is as follows:

$$Z_{in} = R - jX = Z_0 \frac{Z_L + jZ_0 tan\theta}{Z_0 + jZ_L tan\theta} \tag{2}$$

The complex Equation (2) is converted into Equation set (3):

$$\begin{cases} R_L tan\theta - Z_0 X_L G_0 - Z_0^2 tan\theta G_0 - Z_0 R_L B_L = 0 \\ Z_0 - X_L tan\theta - Z_0 R_L G_0 + Z_0 X_L B_L + Z_0^2 tan\theta B_L = 0 \end{cases} \tag{3}$$

As shown in Figure 2, the characteristic impedance of a series microstrip line TL1 is $Z_0 = 50\ \Omega$. The microstrip line TL1 will make the impedance of the first frequency band $f_1$ is converted to a resistance circle with a normalized impedance of 1.

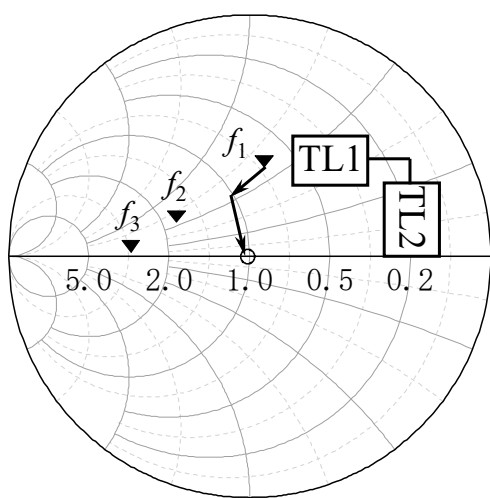

**Figure 2.** The matching circuit for the first frequency band.

The characteristic impedance of a microstrip line is $Z_0 = 50\ \Omega$. The series microstrip line TL1 of transforms the impedance of the first frequency band $f_1$ into the real part of the admittance $G_0 = 0.02$. Then MATLAB software is used to quickly solve the two unknown variables $\theta_1$ and $B_A(f_1)$ of Equation (3). When $B_A(f_1)$ is given. At node A in Figure 1, the microstrip line TL2 of the parallel microstrip branch will cancel the susceptance required by the first frequency band $f_1$. The characteristic impedance of the short transversal of the parallel microstrip branch TL2 is chosen as $Z_0 = 50\ \Omega$. We can find $\theta_2$, as shown in Equation (4):

$$\begin{cases} jB_A(f_1) = jZ_0tan\theta_2 \ \ short \\ -jB_A(f_1) = -jZ_0cot\theta_2 \ \ open \end{cases} \tag{4}$$

Thus, the impedance matching of the first frequency band is completed. At node A in Figure 1, the impedance of the second frequency band $f_2$ and the third frequency band $f_3$ are transformed to $Z_{L2} = R_4 + jX_4$, $Z_{L3} = R_6 + jX_6$. After this, impedance matching for the second frequency band $f_2$ will be achieved.

### 2.2. The Second Band Matching Circuit Design

The impedance value of the second frequency band $f_2$ has been given by $Z_{L2} = R_2 + jX_2$ transform into $Z_{L2} = R_4 + jX_4$. In the same way, A microstrip line TL3, whose characteristic impedance value is $Z_0 = 50\ \Omega$. Is first connected in series. The impedance of the second frequency band $f_2$ is transformed into the real part of the admittance, $G_0 = 0.02$. Then, MATLAB software is used to quickly solve the two unknown variables $\theta_3$ and $B_H(f_2)$ by Equation (3).

As shown in Figure 3, the characteristic impedance of the first microstrip line TL4 is selected as $Z_0 = 50\ \Omega$, and the electrical length is 90 degrees. The second section of microstrip line TL5 is selected as the open-circuit short microstrip line, whose characteristic impedance value is $Z_0 = 50\ \Omega$, and the electrical length is 90 degrees. The electric lengths of the two microstrip lines TL4 and TL5 are based on the first frequency band $f_1$. According to the input impedance of the open circuit, the microstrip line is 0, and the input impedance of the short circuit microstrip line is $\infty$. You can get $Y_H(f_1) = \infty$, at node H in Figure 1, the

susceptance of the second frequency band $f_2$ is canceled as needed $Y_H(f_2)$, The desired load impedance $Z_C(f_2)$ can be obtained by Equation (5):

$$Z_H(f_2) = Z_0 \frac{Z_c(f_2) + jZ_0 tan\left(90\frac{f_2}{f_1}\right)}{Z_0 + jZ_c(f_2) tan\left(90\frac{f_2}{f_1}\right)} \qquad (5)$$

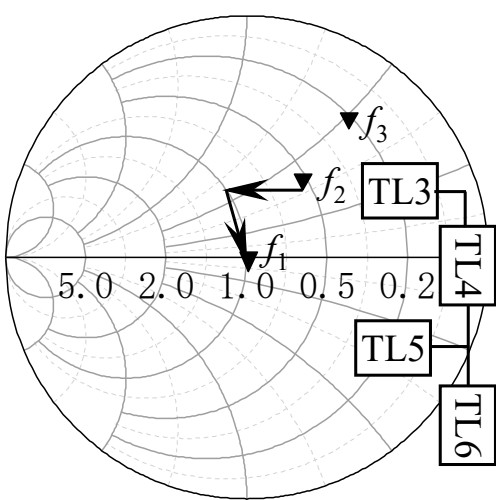

**Figure 3.** The matching circuit for the second frequency band.

When $Z_C(f_2)$ is obtained, the input impedance $Z_{C1}(f_2)$ of open microstrip line TL5 is obtained by Equation (6):

$$Z_{c1}(f_2) = -jcot\left(90\frac{f_2}{f_1}\right) \qquad (6)$$

The desired value $Y_{C2}(f_2)$ of the last parallel microstrip line TL6 can be obtained from Equation (7)

$$Y_{c2}(f_2) = Y_c(f_2) - Y_{c1}(f_2) \qquad (7)$$

The characteristic impedance of the last section of parallel microstrip line TL6 is $Z_0 = 50 \ \Omega$, and the electric length of the parallel microstrip line $\theta_4@f_2$ can be obtained through Equation (8)

$$\begin{cases} \theta_4 = arctan(\frac{B_{c2}(f_2)}{Z_0}) \ \ short \\ \theta_4 = arccot(\frac{B_{c2}(f_2)}{Z_0}) \ \ open \end{cases} \qquad (8)$$

Thus, the impedance value of the third frequency band $f_3$ is transformed to $Z_{L3} = R_6 + jX_6$. Next, the impedance matching of the third frequency band will be implemented.

### 2.3. The Third Band Matching Circuit Design

As shown in Figure 4, the impedance of the third frequency band $f_3$ is matching, the impedance of the first frequency band $f_1$ and the second frequency band $f_2$ has been matched to 50 $\Omega$. In this part, the impedance value of the third frequency band $f_3$ is transformed to $Z_{L3} = R_6 + jX_6$. Similarly, a microstrip line TL7 with a characteristic impedance value of $Z_0 = 50 \ \Omega$ is connected in series and the real part $R_6$ of the impedance of the third frequency band $f_3$ is transformed into $G_0 = 0.02$. The first frequency band $f_1$ and the second band $f_2$ will not be affected. (Because the characteristic impedance value of the series microstrip line is $Z_0 = 50 \ \Omega$). The impedance of the third frequency band $f_3$ transforms to $Y_{L3} = 0.02 + jB_N(f_3)$, a $\pi$-type microstrip line structure TL8, TL9, TL10, TL11, TL12 is connected in parallel to cancel the susceptance value of the third frequency band $f_3$. Equivalent circuits in both the first and second frequency bands are open circuits. The parameters of the microstrip line of the whole matching circuit are obtained.

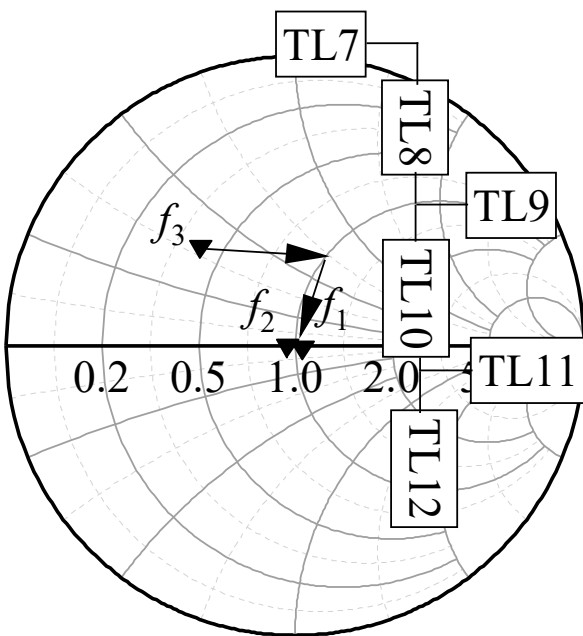

**Figure 4.** The matching circuit for the third frequency band.

The approach designed in this article has some similarities with the Class-F power amplifier. In terms of harmonic control, The Class-F power amplifier matching method is a special case. It is no universality; this article proposes a point-to-point matching method. The match method is universality.

### 3. Design of Bias Circuit

The power amplifier consists of a transistor, an input-matching circuit, an output-matching circuit, and a bias circuit. The bias circuit protects the DC power supply by blocking the RF signal. The design methods of bias circuits are quarter wavelength microstrip lines or broadband bias. In this paper, a quarter wavelength microstrip line is used to block three frequency bands at the same time. In this way, the input impedance of the bias circuit in all three frequency bands needs to be equivalent to the open circuit. That is, the input impedance goes to infinity. As shown in Figure 5, the biased circuit structure can achieve high impedance values in three frequency bands simultaneously. The characteristic impedance $Z_0$ of the first microstrip line is 50 $\Omega$. Based on the first frequency band $f_1$, the electrical length $\theta$ of the microstrip line is 90 degrees. The characteristic impedance value $Z_0$ of the second part of the microstrip line is 50 $\Omega$. Based on the first frequency band $f_1$, the electrical length $\theta$ of the microstrip line is also 90 degrees, but the load is an open circuit. In this way, the first open equivalent circuit of the frequency band is realized. The input impedance value of the third part can be obtained from Equations (6) and (7), and the characteristic impedance value $Z_0$ of the third part of the microstrip line is 50 $\Omega$. Based on the second frequency band $f_2$, the electric length $\theta$ of the microstrip line can be obtained by Equation (8). The bias circuit is the DC input channel, which has a great influence on the efficiency of the power amplifier. Moreover, a parallel grounding capacitor is required to prevent self-excited oscillation of the circuit.

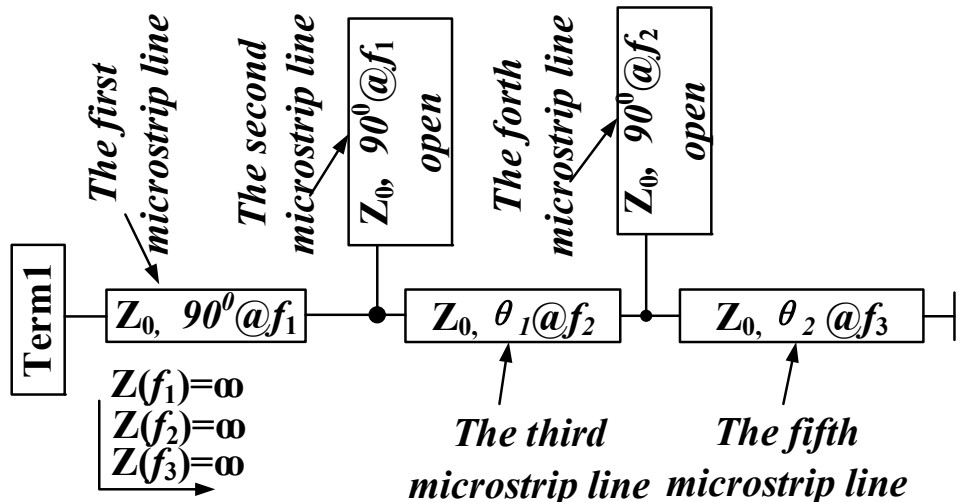

**Figure 5.** Biasing circuit network structure.

## 4. Design and Simulation

The design of gallium nitride (GaN) transistor CGH40100F from Cree company is used to realize the concurrent tri-band high-efficiency power amplifier. The optimal source impedance and load impedance of the transistor in the three frequency bands of 2.2 GHz, 2.8 GHz, and 3.5 GHz are tested by the load pull and source pull of Advance Design Software (ADS), as shown in Table 1.

**Table 1.** Load impedance and source impedance.

| Frequency | Source Impedance | Load Impedance |
|-----------|------------------|----------------|
| 2.2 GHz | $6.6 - j*4.6$ | $6.7 + j*2.0$ |
| 2.6 GHz | $3.2 - j*4.7$ | $19 + j*9.2$ |
| 3.5 GHz | $3.1 - j*13.3$ | $14.6 + j*7.4$ |

Figure 6 is the circuit structure diagram of the overall power amplifier. The input-matching circuit and the output-matching circuit adopt the same circuit structure, and the drain bias circuit and the gate bias circuit adopt the same circuit structure. The impedance values of both ports are 50 Ω. According to the proposed concurrent tri-band matching circuit, the output impedance and load impedance of the three frequency bands can be matched simultaneously. When the transistor gate voltage is set to 2.9 V, the drain voltage is set to 28 V. Through the test of software ADS, S parameters are obtained. As shown in Figure 7, The $S_{11}$ is less than $-10$ dB in the three frequency bands of 2.2/2.6/3.5 GHz and about 0 dB~0.5 dB in the other frequency bands. The $S_{21}$ reached around 15 dB in all three frequency bands. This shows that the reflection loss between the impedance matching of the three frequency bands is small, and the isolation between the bands is high.

Figure 7 shows the s parameters simulation results of concurrent tri-band RF power amplifiers in the frequency bands of 2.2 GHz, 2.6 GHz, and 3.5 GHz. $S_{21}$ shows the highest point at 2.2 GHz, 2.6 GHz, and 3.5 GHz. The $S_{11}$ shows the lowest point at 2.2 GHz, 2.6 GHz, and 3.5 GHz. This illustrates that the designed power amplifier has suitable tri-band characteristics.

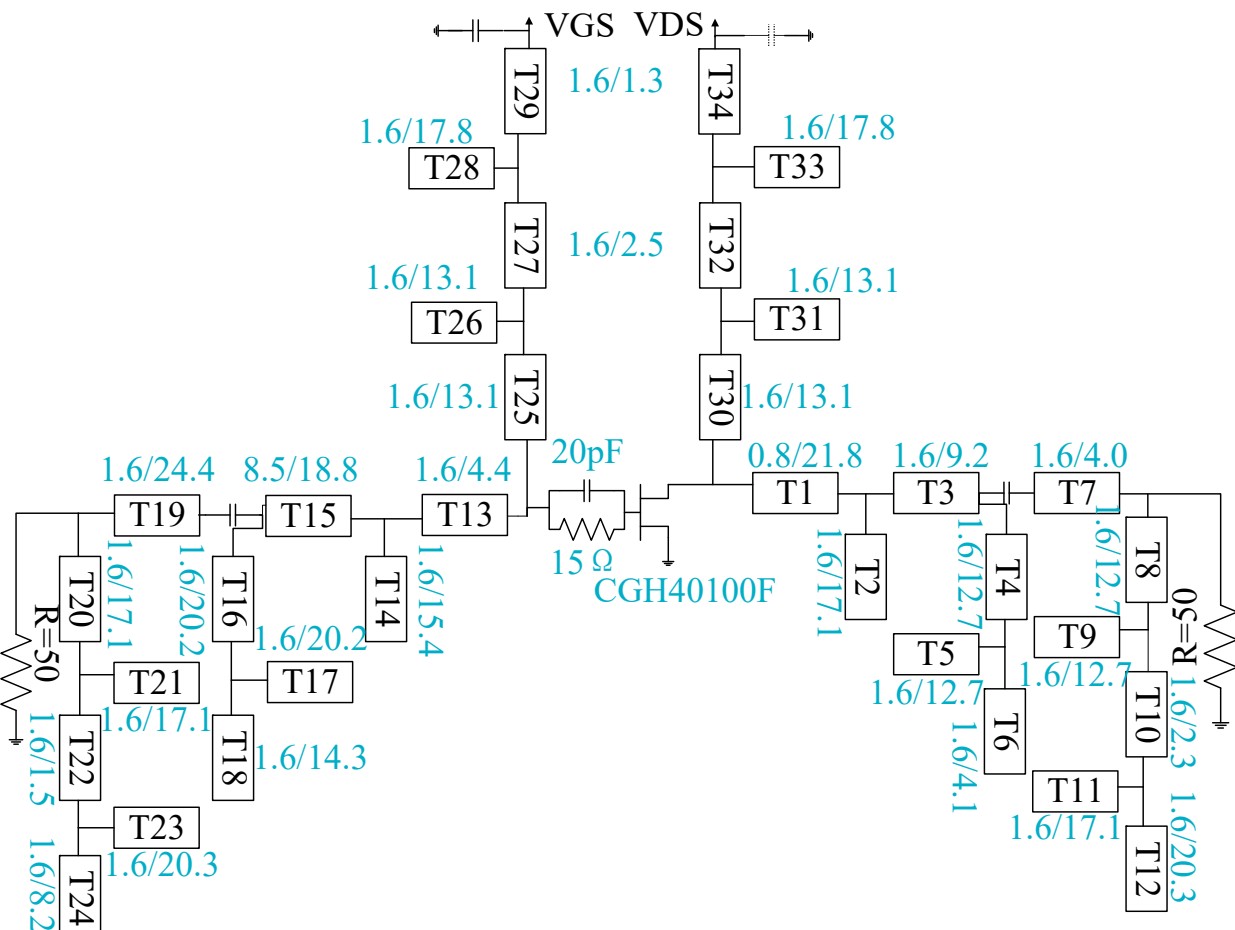

**Figure 6.** Overall power amplifier topology. Dimension: width/length (mm).

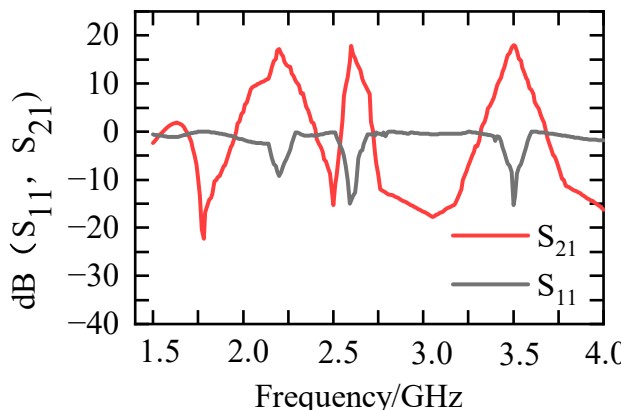

**Figure 7.** Test results of s parameters.

The output power, power-added efficiency, and gain results at the three frequency bands of 2.2/2.6/3.5 GHz are shown in Figure 8. At 2.2 GHz, the input power increases from 10 dBm to 30 dBm, and the output power and power-added efficiency increase with the input power. The peak output power reaches more than 40 dBm, the power-added efficiency reaches between 50% and 63%, and the gain is about 11–13 dBm. At 2.6 GHz, the input power increases from 11 dBm to 30 dBm, and the output power and power-added efficiency increase with the input power. The peak output power reaches more than 40 dBm, and the power-added efficiency reaches between 65% and 70%. The gain is about 12 to 13 dBm. At 3.5 GHz, the input power increases from 10 dBm to 30 dBm, and the output

power and power-added efficiency increase with the input power. The peak output power reaches more than 40 dBm, the power-added efficiency reaches between 55% and 60%, and the gain is about 14–15 dBm. The performance indicators of the three frequency bands have achieved suitable results. This also confirms the feasibility of the designed concurrent tri-band high-efficiency power amplifier.

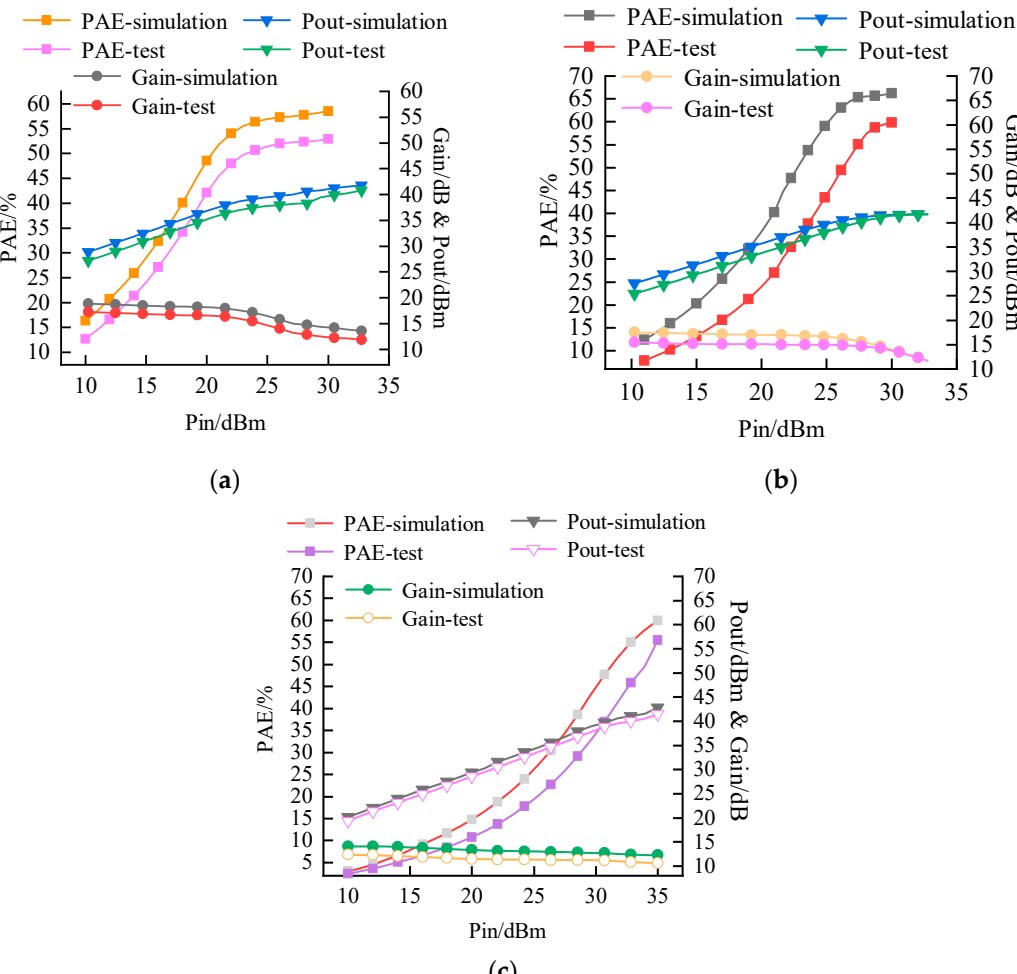

**Figure 8.** Simulation results and test results of power-additional effect, output power, and gain. (**a**) at 2.2 GHz simulation and test results. (**b**) at 2.6 GHz simulation and test results. (**c**) at 3.5 GHz simulation and test results.

Shown in Figure 9 are the IM3 test results. The absolute value of IM3 at low input power is greater than 70, but with the increase in input power, the absolute value of IM3 becomes smaller, and the nonlinearity is more serious. In RF or microwave multi-carrier communication systems, the third-order intermodulation is an important index used to measure the nonlinearity of RF devices, and its size is expressed by the ratio between the intermodulation product and the main output signal, the unit is dBc. The greater the absolute value of the third-order intermodulation index, the better. The larger the value, the smaller the intermodulation product is relative to the main signal and the smaller the interference effect on the system. Any semiconductor device has certain nonlinearity, especially in the case of large signal input. The nonlinearity will be more obvious. Because amplifiers have a certain gain, this means that amplifiers are more nonlinear than other semiconductor devices, which is why we pay special attention to amplifier nonlinearity in practice. In fact, in addition to the third-order term will be generated, the fifth, seventh,

and other odd higher-order terms can also be generated, but the higher the order, the less contribution.

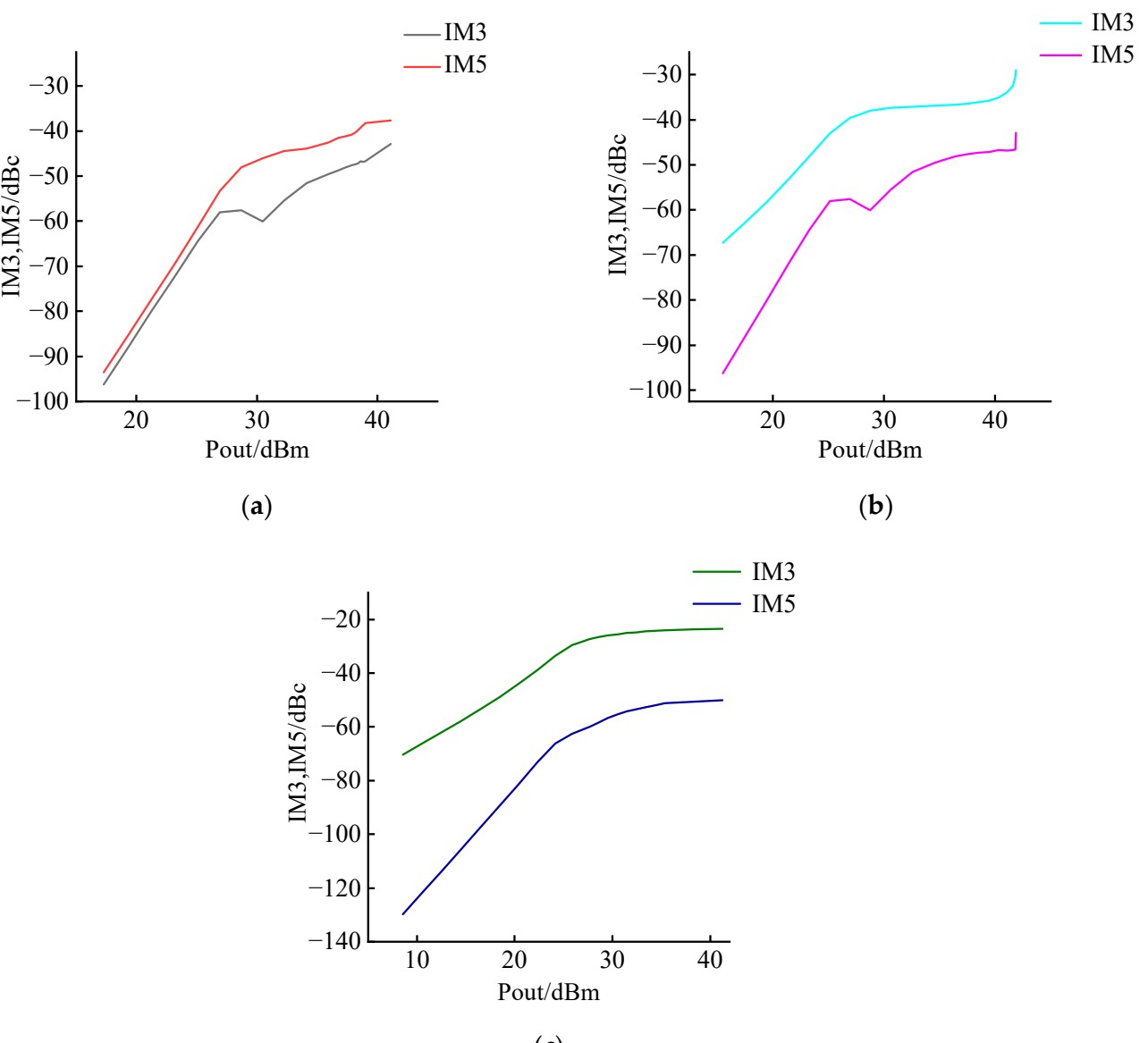

**Figure 9.** Third-order intermodulation distortion and fifth-order intermodulation distortion test results. (**a**) at 2.2 GHz simulation results. (**b**) at 2.6 GHz simulation results. (**c**) at 3.5 GHz simulation results.

As shown in Figure 10, The substrate material of Rogers5880 is adopted, and the dielectric substrate with a thickness of 0.508 mm and a dielectric constant of 3.66 was selected for the physical production of the power amplifier. The gate bias voltage is set to −2.9 V, and the drain bias voltage is set to 28 V. The power amplifier operates in the class AB offset state. The large signal simulation of the power amplifier uses single-tone CW to test the output power, power-added efficiency, and gain at three operating frequencies of 2.2/2.6/3.5 GHz. The fixed input power is 30 dBm, and the scanning signal frequency range is 2.0–4.0 GHz. The characteristics of the large signal performance of the power amplifier with frequency variation are tested.

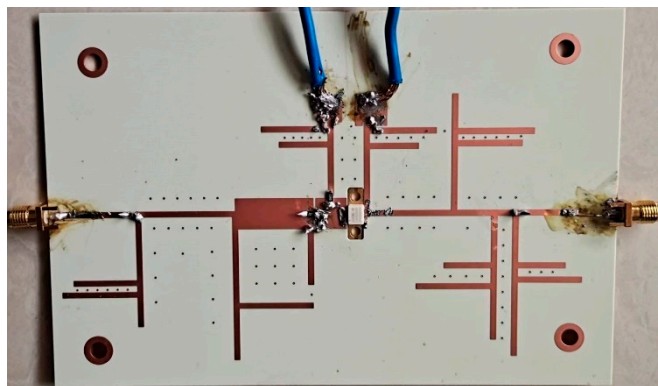

**Figure 10.** Picture of power amplifier PCB board.

The test results and simulation results are shown in Figure 11. The power-added efficiency has a local maximum of 2.2 GHz, 2.6 GHz, and 3.5 GHz. The output power also reaches about 40 dBm, and the gain is about 10 dB. It shows that the designed power amplifier has suitable concurrent tri-band characteristics. The curves of power-added efficiency (PAE), output power (Pout), and gain as a function of input power (RFpower) are shown in Figure 11. According to Figure 11, the measured peak efficiency (PAE) of the power amplifier at the three frequency points is 61/67.8/56%, the maximum output power is 41/42/40.5 dBm, and the large signal gain is about 11 dB. There is a gap between realistic welding and ideal simulation. Due to the difference between the transistor model and the actual device, and the parasitic parameters in the welding of components such as resistors and capacitors, the measured performance is slightly worse than the simulation performance. Compared with the ideal environment of software simulation, the uncertainty of the packaging model and the fact that the 8 mm welding microstrip line is reserved for convenient welding will lead to a change in the measured results.

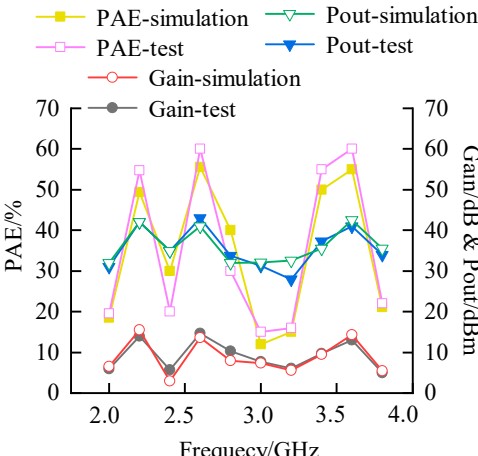

**Figure 11.** Simulation and test results of power addition, power output, and gain at 2.0 GHz to 4.0 GHz.

Table 2 compares the parameters of the concurrent tri-band power amplifier designed in this article with the published literature on the concurrent dual-band power amplifier. The same type of power amplifier, CGH40100F, is used in the literature. It can be seen from Table 2 that the performance of the proposed tri-band power amplifier is excellent. Other are concurrent dual-frequency power amplifiers. This is a significant advantage.

**Table 2.** Performance comparison of same type power amplifiers.

| Ref. | Frequency/GHz | Output Power/dBm | PAE/% | Gain/dB |
|---|---|---|---|---|
| [4] | 0.73/1.65/2.67/3.57 | 42.3/42.4/42.5/41.7 | 54.6/44.6/58.9/51.4 | 10/10.2/10.3/10.1 |
| [5] | 1.96/3.5 | 40.19/39.60 | 56.94/49.51 | 10.7/10.8 |
| [7] | 0.8/1.9 | 41.01/40.03 | 65/68 | 13.0/12.5 |
| [8] | 1.0/2.3 | 41.6/42.1 | 72.4/74.1 | 11.2/11.6 |
| This work | 2.2/2.6/3.5 | 40.05/41.50/41.02 | 55/69/65 | 11.0/14.0/6.5 |

The output power of the concurrent tri-band power amplifier in the three frequency bands of 2.2/2.6/3.5 GHz has reached more than 40 dBm, which has reached the indicators parameters given in the data sheet of the CGH40100F transistor. The power-added efficiency reaches more than 50%, and the efficiency is slightly lower than other ones. There is no harmonic control performed. To achieve the mutual isolation between the three frequency bands, the use of an open branch results in a large volume and also reduces the efficiency gain. However, the proposed concurrent tri-band power amplifier meets the basic requirements.

## 5. Conclusions

This article presents a design method for a concurrent tri-band high-efficiency power amplifier for a concurrent multiband transmission system. Firstly, the matching circuit of a concurrent tri-band high-efficiency power amplifier is analyzed in detail, and the isolation between three frequency bands and the matching of impedance values of three frequency bands are realized by using the principle of impedance compensation and the equation of matching circuit and bias circuit of the concurrent tri-band band are derived. Then, the proposed matching circuit structure achieves accurate impedance matching of the three frequency bands. Moreover, the design of a concurrent quad-band power amplifier or even a concurrent five-band power amplifier can be carried out on this circuit. The analytical equation of the design parameters of the matching circuit and the detailed design process are derived. Finally, a power amplifier working at 2.2/2.6/3.5 GHz is designed to verify the feasibility of the scheme. The test results illustrate that the saturated output power of the power amplifier reaches more than 10 W in the frequency bands of 2.2 GHz, 2.6 GHz, and 3.5 GHz, the peak efficiency is about 55~68%, and the gain is about 11 dB. It shows a relatively suitable performance indicator. The structure of the power amplifier is a concurrent tri-band power amplifier, which is an effective solution to realize the efficient operation of the RF power amplifier in three frequency bands simultaneously. The Class-F power amplifiers need to control several harmonics. This is the dual-band/tri-band match circuit. However, there are some differences in this approach. The Class-F power amplifier controls the harmonic by open and short circuits. It is simply achieved by a quarter microstrip line and a half microstrip line. While the impedance is not open and short, what we need becomes more different and complex. So, my article proposes an approach to complete the three-impedance match in the three frequency bands. The idea and structure are the same, but my work has been more deeply processed in the three frequency bands. It is more applicable to most situations.

**Author Contributions:** Conceptualization, M.G. and G.X.; methodology, M.G.; software, G.X.; validation, M.G., J.N. and G.X.; formal analysis, G.X.; investigation, M.G.; resources, G.X.; data curation, M.G.; writing—original draft preparation, G.X.; writing—review and editing, G.X.; visualization, J.N.; supervision, M.G.; project administration, M.G.; funding acquisition, M.G. All authors have read and agreed to the published version of the manuscript.

**Funding:** This work was funded by the 1. Applied Basic Research Project of Liaoning Province (2022JH2/101300275) 2. National Foundation: Power Amplifier Design Modeling and predistortion research for cognitive Radio Systems under compressed sensing Framework (61701211) 3.

Future-oriented Research on Wireless Reconfigurable Intelligent RADIO Module and neural network Modeling (61971210).

**Data Availability Statement:** The data that support the findings of this study are available from the corresponding author, [Gaoyang Xu], upon reasonable request.

**Conflicts of Interest:** The authors declare no conflict of interest.

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
