# Peer review of "Design of Concurrent Tri-Band High-Efficiency Power Amplifier Based on Wireless Applications"

_electronics, doi:10.3390/electronics11213544_

Round 1
Reviewer 1 Report (Previous Reviewer 3)
The paper presents a 3-band amplifier with distributed component impedance matching circuits at the input and output.
Suggestions for improvements:
1) Number and units must have a space between them;
2) figure 9b has a yellow trace which is barely visible. must be changed to something more vivid.
I'd suggest accepting the paper after the above mentioned minor changes are done.
Author Response
Point 1: Number and units must have a space between them;
Response 1: I will correct it carefully
Point 2: figure 9b has a yellow trace which is barely visible. must be changed to something more vivid.
Response 2: I will change the color.
Thank you for your comments and your accept.

Reviewer 2 Report (New Reviewer)
According to the Authors, this paper elaborates on:
a) Introduction, Page 2 “.. proposes a concurrent three-band high efficiency power amplifier. Impedance compensation principle is adopted. The input, output and bias circuits of concurrent three band high-efficiency power amplifier are designed, and the input and output impedance matching of three bands and the high isolation between each band are realized”
b) Section 5, Page 5, “In this paper, a quarter wavelength microstrip line is used 210 to block three frequency bands at the same time. In this way, the input impedance of the 211 bias circuit in all three frequency bands needs to be equivalent to open circuit.”
Indeed, the Authors herein were able design the matching circuits for an amplifier operating concurrently in three different frequency bands. However,
1) The same matching approach is repeated multiple times, making the reading boring and distracting the Readers attention. The approach should be explained only once and then to be applied as many times as needed.
2) A very similar matching approach is well established in the design of “Class-F power amplifiers” or “Inverse class-F power amplifiers”, were several harmonics must be matched to present specific impedance (imaginary, zero or short circuited or infinite open circuited). A plethora of related publications can be found by a simple search, e.g. [R-1]. Thus, the Authors should clarify, what are the common approach shared with these techniques and where these are different from the proposed one.
[R-1] Hemin Wu, Kelvin Yuk, Can Cui, George R. Branner, “High Power Class F GaN HEMT Power Amplifier in L band for Global Positioning Systems Application”, April 2018, DOI: 10.1109/WAMICON.2018.8363920, Conference: 2018 IEEE 19th Wireless and Microwave Technology Conference (WAMICON), At: Sand Key, Florida, USA
[R-2] Mahya Parnianchi, “ High-efficiency class-F Power amplifier with a new design of input matching network”,
INTERNATIONAL JOURNAL OF CIRCUITS, SYSTEMS AND SIGNAL PROCESSING DOI: 10.46300/9106.2022.16.106
[R-3] Fei Yang , Hongxi Yu, Jun Li, Chao Guo, Sen Yan, Xiaoming Chen, Anxue Zhang, and Zhonghe Jin, “A Class-F Based Power Amplifier with Optimized Efficiency in Triple-Band”, Electronics 2022, 11, 310. https://doi.org/10.3390/electronics11030310
3) In particular reference [R-3] presents a very similar approach in Fig,4 therein. Please, clearly justify the novelty of the present approach with respect to [R-3].
4) Section 6, presents the classical conjugate matching as something new which is proposed herein. This is completely misleading. The Authors should avoid such misconceptions.
5) Small signal S-parameters are utilized herein (e.g. p.10, line 210) throughout the paper for the design of a power amplifier. However, it is well established that large signal S-parameters, along with large signal sources and Harmonic balance simulation, should be utilize in the HPA design.
6) At the beginning of section 2 the term “load traction” is used, but this is well established as “load pull”. Notably, load pull technique is necessary because of the non-linearity of the circuit related to the presence of large signals.
Overall, this paper should be placed in the context of “large signal device modelling and design”, while its relation to Class-F power amplifiers” or “Inverse class-F power amplifiers” matching circuits, should be clarified, along with any possible advantages.
Author Response
Point 1: The same matching approach is repeated multiple times, making the reading boring and distracting the Readers attention. The approach should be explained only once and then to be applied as many times as needed.
Response 1: I feel sorry, it is my logics problem. I do not have the Strong overview ability. But in the beginning while I write the paper. I want to display the formula derivation of the concurrent tri-band match network. While the formula derivation is complex and boring. The first band match network and the second match network are something different in the last process. So, I repeat the method again, I will continue Simplify my language.
Point 2: A very similar matching approach is well established in the design of “Class-F power amplifiers” or “Inverse class-F power amplifiers”, where several harmonics must be matched to present specific impedance (imaginary, zero or short circuited or infinite open circuited). A plethora of related publications can be found by a simple search, e.g. [R-1]. Thus, the Authors should clarify, what are the common approach shared with these techniques and where these are different from the proposed one.
[R-1] Hemin Wu,Kelvin Yuk,Can Cui,George R. Branner, “High Power Class F GaN HEMT Power Amplifier in L band for Global Positioning Systems Application”, April 2018, DOI:10.1109/WAMICON.2018.8363920, Conference: 2018 IEEE 19th Wireless and Microwave Technology Conference (WAMICON), At: Sand Key, Florida, USA
Response 2: The class-F power amplifiers need control several harmonics. This is the dual band/tri-band match circuit. But there are some differences from my approach. As we all know, the class-F power amplifier controls the harmonic by open and short circuit. It is simple achieve by the A quarter microstrip line and a half microstrip line. While the impedance is not open and short what we need that becomes more different and complex. so, my article proposes a approach to complete the three impedances match in the three frequencies. The idea and structure are same, but my work have more deeply processing in the three frequencies. It is more applicable to most situations.
Point 3: In particular reference [R-3] presents a very similar approach in Fig,4 therein. Please, clearly justify the novelty of the present approach with respect to [R-3].
[R-3] Fei Yang, Hongxi Yu, Jun Li, Chao Guo, Sen Yan, Xiaoming Chen, Anxue Zhang, and Zhonghe Jin, “A Class-F Based Power Amplifier with Optimized Efficiency in Triple-Band”, Electronics2022, 11, 310. https://doi.org/10.3390/electronics11030310
Response 3: I didn't read the article before I wrote it. Because the article was published very late. When I finish reading this article. i have some conclusion.
- in the [R-3], Although the topic is concurrent tri-frequency power amplifier, the implementation methods are different.
In the section 2 Design Space of Output Load Termination versus DE.” As discussed in the introduction, it is difficult to realize nine precise impedances matching for the triple-band Class-F mode PA, since the output matching network will be so complex that it cannot be completed”. The author said it difficult to complete nine precise impedances matching, so in the following, the impedance matching of three frequency bands is achieved by using a low-pass filter. This is also in the section 2 “Based on our load termination matching experiences, by using the low-pass topology, the second and third harmonic impedances can be theoretically arranged within the edge of the Smith chart but are difficult to locate in the specific points”.
So, I think the method used in this article to implement concurrent tri-band is different from the one proposed in my article. The authors use low-pass filters to implement concurrent tri-frequency power amplifiers, while I use a point-to-point matching method
Point 4: Section 6, presents the classical conjugate matching as something new which is proposed herein. This is completely misleading. The Authors should avoid such misconceptions.
Response 4: I will delete it.
Point 5: Small signal S-parameters are utilized herein (e.g., p.10, line 210) throughout the paper for the design of a power amplifier. However, it is well established that large signal S-parameters, along with large signal sources and Harmonic balance simulation, should be utilize in the HPA design.
Response 5: I correct it right away.
Point 6: At the beginning of section 2 the term “load traction” is used, but this is well established as “load pull”. Notably, load pull technique is necessary because of the non-linearity of the circuit related to the presence of large signals.
Response 6: Sorry, this is my mistake carefully less, I will correct it.
Point 7: Overall, this paper should be placed in the context of “large signal device modelling and design”, while its relation to Class-F power amplifiers” or “Inverse class-F power amplifiers” matching circuits, should be clarified, along with any possible advantages.
Response 7: I'm sorry I forgot to consider one important factor- “device modeling”, To be honest, I do not do the research of the Class-F power amplifier deeply. In terms of harmonic control, I think Class-F matching method is a special case. It is no universality; I propose the match method is universality.

Reviewer 3 Report (Previous Reviewer 2)
1. Please show the equation of the first band matching circuit design.
2. Please show the equation of the second band matching circuit design.
3. Please show the equation of the third band matching circuit design.
4. Please show the equation of the 5. Design of bias circuit.
5. The author’s manuscript said “The peak output power reaches more than 40dBm, the power added efficiency reaches between 50% and 63%, and the gain is about 11-13dbm.”
Please show the equation of efficiency.
6. Please show the equation of the Fig. 7.
7. Please show the equation of Fig. 8.
8. Please show the equation of Fig. 9.
9. Please show the equation of Fig. 10.
10. Please show the equation of Fig. 12.
11. The author’s manuscript said “The power added efficiency reaches more than 50%, and the efficiency is slightly lower than other ones, because the matching circuit has more open branches, resulting in large losses.”
Please show the equation of the efficiency, which can go the calculation.
12. Please show the figure of the other reference [?] of Fig. 7 Fig. 8, Fig. 9, Fig. 10, Fig. 12.
Round 2
Reviewer 2 Report (New Reviewer)
Dear Authors,
Thank you for trying to address my comments in your separate reply. However, the corresponding changes should be included in the paper. Explicitly:
1. 1st comment "repeating the matching approach multiple times": You should describe the methodology only once in a detailed form. In turn you may apply this for the other cases. The paper must be changed accordingly.
2. 2nd and 3rd comments: "Similarity of the proposed approach to the one followed in the design of class-F HPAs". The similarities and differences as well as the related advantages or novelty must be include in the paper.
Author Response
Response to Reviewer 2 Comments
Point 1: 1st comment "repeating the matching approach multiple times": You should describe the methodology only once in a detailed form. In turn you may apply this for the other cases. The paper must be changed accordingly.
Response 1: I'll change it right away
Point 2: 2nd and 3rd comments: "Similarity of the proposed approach to the one followed in the design of class-F HPAs". The similarities and differences as well as the related advantages or novelty must be included in the paper.
Response 2: I will add these points.

Reviewer 3 Report (Previous Reviewer 2)
1. It is good enough to publish the Electronics.
Author Response
Response to Reviewer 3 Comments
Point 1: It is good enough to publish the Electronics.
Response 1: Thank you for your comments and recognition

Round 3
Reviewer 2 Report (New Reviewer)
Dear Authors,
Thank you for addressing my comments, however your paper needs extensive language editing.
This manuscript is a resubmission of an earlier submission. The following is a list of the peer review reports and author responses from that submission.
Round 1
Reviewer 1 Report
While the topic is of some interest, it is not timely and I found it very difficult to follow the paper. The description is not clear to me. Similar aspects have been covered in other published papers and it is not clear to me what additional is provided by the submitted paper. I would encourage the authors to revisit their work in such a direction. The authors need to explain the added value of their paper.
Reviewer 2 Report
1. Please show the equation of the first band matching circuit design.
2. Please show the equation of the second band matching circuit design.
3. Please show the equation of the third band matching circuit design.
4. Please show the equation of the 5. Design of bias circuit.
5. The author’s manuscript said “The peak output power reaches more than 40dBm, the power added efficiency reaches between 50% and 63%, and the gain is about 11-13dbm.”
Please show the equation of efficiency.
6. Please show the equation of the Fig. 7.
7. Please show the equation of Fig. 8.
8. Please show the equation of Fig. 9.
9. Please show the equation of Fig. 10.
10. Please show the equation of Fig. 12.
11. The author’s manuscript said “The power added efficiency reaches more than 50%, and the efficiency is slightly lower than other ones, because the matching circuit has more open branches, resulting in large losses.”
Please show the equation of the efficiency, which can go into the calculation.
12 Please show the figure of the other reference [?] of Fig. 7 Fig. 8, Fig. 9, Fig. 10, and Fig. 12.
Reviewer 3 Report
1) Some parts of the text contain Figure 1, some FIG. 2. This has to be unified under MDPI template requirements.
2) A lot of text formatting errors are present in page 2. Can't mark them out, as the provided template doesn't have line numbers.
3) Fig.7 I guess the S11 and S21 colours are wrong, as if S11 was to be more than 0, you'd have an oscillator.
4) formatting errors su chas ghz (not GHz) are present on page 8
5) What is the difference in your design approach comapred to others? This would be preferable to distinguish, because currently nothing is mentioned, just the process of designing is shown.
6) Based on table 2 - why is the PAE in [8]/[4] on a lower frequency range around 70%, and your solution only provides 55%? BOth of these works don't use open-stubs?
7) " Generally speaking, the performance of concurrent tri-band power amplifier meets the basic requirements" - what are the requirements? how would the PA cope with a modulated signal, would this configuration pass ACPR requirements for, lets say, 4G or 5G?
